# Measuring Uncertainty Analysis of the New Leveling Staff Calibration System

**DOI:** 10.3390/s23146358

**Published:** 2023-07-13

**Authors:** Sergej Baričević, Tomislav Staroveški, Đuro Barković, Mladen Zrinjski

**Affiliations:** 1Faculty of Geodesy, University of Zagreb, Kačićeva 26, 10000 Zagreb, Croatia; 2Faculty of Mechanical Engineering and Naval Architecture, University of Zagreb, Ivana Lučića 5, 10002 Zagreb, Croatia

**Keywords:** precise leveling staffs, horizontal comparator, accuracy assessment, calibration accuracy

## Abstract

Besides precise levels, precise leveling staffs are a crucial part of the measuring equipment when carrying out geodetic (geometric) leveling measurements. The leveling staffs define the scale of the height reference system, so it is important to calibrate them periodically and when necessary. This paper shortly describes the development of the new method of calibrating leveling staffs in the Laboratory for Measurements and Measuring Technique of the Faculty of Geodesy, University of Zagreb. The existing horizontal comparator was upgraded by installing a servo-motorized positioning drive with a mounted CCD camera and telecentric lens that is used to record graduations of the leveling staffs. The software was developed to support the management of the comparator system, as well as for the analysis and processing of images and measurement data and, most importantly, giving the result in the form of a calibration report. The main subject of this paper is a detailed assessment of the measurement uncertainty of determining the position of the edges of the graduation lines and determining the scale of precise centimeter and coded leveling staffs. The estimates were confirmed by experimental measurements.

## 1. Introduction

Precise geodetic (geometric) leveling is the most precise method to determine the height difference between two points. With the usage of functional and calibrated equipment and by application of the right methods and procedures, accuracy up to the level of a hundredth of a millimeter for individual measurement or up to 0.2 mm/km with double run leveling can be achieved [1]. The crucial part of leveling measurement equipment is a set consisting of precise level (optical or digital) and leveling staff or staffs (analog or coded). Although leveling staffs are considered auxiliary equipment, they are equally important as the levels themselves. Leveling without one or another cannot be conducted. Due to time and usage, every instrument’s performance may deteriorate. To check the properties of the equipment, some standards recommend periodical tests of levels and staffs [2,3]. There are two principles of evaluating the quality of equipment: calibration of leveling set and calibration of leveling staff only. Prior to development of digital levels, testing and calibration of a leveling set was usually conducted by a field test. Individual laboratory checkups of levels were conducted by a mechanical and optical inspection and of staffs by calibration in the comparator. With the development of digital levels, the number of laboratory comparators for testing digital leveling increased. In that way, environmental conditions’ influence on the calibration results is minimized. Either way, calibrating the set or only the staff, the result of the comparation is a calibration certificate that allows correction of the raw field leveling data. Both principles may have advantages and disadvantages, but it is up to the user to decide which one is best suited for their leveling purposes. Another condition on deciding what type of calibration to use is that there are no commercial comparators as turnkey solutions. So, comparators are individually constructed by different institutions. Comparator construction depends on more factors, such as calibration type, position of staff, staff reading, distance (or movement) measurement device, and moving parts, which are dependable on available funding (Table 1).

As mentioned, the main classification of leveling equipment comparators is according to calibration subject: leveling set or leveling staff. According to available data, there are more staff comparators worldwide but with an increasing number of leveling set comparators. The only comparator that has the capacity for both calibration types is at the SLAC National Accelerator Laboratory Stanford at the University of Stanford, USA [5,6,7]. According to the available literature, the best known and best in terms of leveling set calibration accuracy is the comparator at the Institute of Engineering Geodesy and Measurement Systems at the Graz University of Technology (TUG) in Austria [8,9,10]. Besides those two, there are some other well-known laboratories with developed calibration systems for leveling sets: Finnish Geodetic Institute (FGI) as the National Standards Laboratory for Free Fall of Acceleration and Length in Masala, Finland (staff comparator turned into set comparator) [11], Department of Geomatics at the Czech Technical University (CTU) in the Czech Republic [12], Department of Surveying and Mapping Malaysia (JUPEM) in Malaysia [12], and the Laboratory of Geodetic Instrumentation (LAIG) of Federal University of Paraná (UFPR) in Brazil [13].

For precise leveling special staffs are used, and International standard ISO 12858-1:2014 [14] specifies requirements in construction, invar scale tape quality, numbering and graduation execution, zero-point error, baseplate (footplate) quality, circular level precision, and calibration of precise leveling staffs. Their body is usually 2 or 3 m long, made in one piece out of aluminum alloy. Graduations are not directly applied on the stuff body but are engraved on a narrow invar tape [15]. The tape is attached to the bottom of the staff, while at the top of the staff invar tape is pre-loaded by a spring with a force of 196 N. Invar is an alloy made out of 64% steel and 36% nickel. It is known for its unique property of an extremely low coefficient of linear thermal expansion (α < 2 × 10^−6^ K^−1^) [16]. Hence, precise leveling staffs are known as invar staffs. Graduations on precise leveling staffs for optical levels are applied on invar tapes in two columns. Individual graduation lines are applied at intervals of 0.5 cm or 1 cm, with a thickness (height) of about 1.5 mm and a width of 6 mm, and can have sharp or rounded edges. The center of each graduation line (centerline between the upper and lower edge) represents the nominal value which is written on the body of the staff next to the graduation line. One of the graduations (left or right) is usually the reference one; that is, its values represent the actual distance from the bottom of the baseplate, while the other graduation is the control one and shifted by a certain constant amount depending on the manufacturer brand. Graduations on the coded staffs are in single column and extend across the entire width of the invar tape and look like barcodes. Staffs with coded graduation are not standardized like those on analog staffs because each manufacturer has patented their own pseudo-barcode pattern, depending on the method of measurement and data processing in the digital level [17]. After assembly, every staff goes through quality control that ensures maximum deviation of each graduation line of ±0.007 mm (±7 µm) and scale deviation of the whole graduation under ±1.2 ppm [15].

With the development of technology, instrumentation has reached a certain level where reducing measurement uncertainty is no longer possible by using higher quality instruments. Therefore, research is focused on detecting and modeling the effects of systematic errors to achieve prescribed criteria for measurement uncertainty. One of the systematic effects on geodetic leveling measurements is due to imperfections in the measuring equipment, namely the leveling staffs. In Table 2 are listed and described the possible impacts of systematic errors of precise leveling staffs on high-accuracy leveling measurements, as well as methods and techniques for minimizing or eliminating them. As can be seen from Table 2, the error due to bending and torsion of staff has been eliminated by using an aluminum alloy for staff body production. The error due to non-verticality of staff can be eliminated by regular testing and adjustment of staff’s circular bubble level. In case the bottom of the baseplate is bumpy (rough) or is not perpendicular with respect to staff axis (that in the working position should be vertical in space), placing the staff on a benchmark with different contact points can affect the measurements with great magnitude. Such an error can be eliminated by using a baseplate ring or baseplate attachment with a hard tip that assures a constant contact point when placing the staff on the benchmark. Zero error of one staff or zero error for a pair of staffs is introduced to measurements when the bottom or bottoms of the baseplates do not coincide with the zero graduation line. That influence can be eliminated by using the same staff or having an even number of setups between two benchmarks when using a pair of staffs. There are cases when conducting leveling measurements, especially in urban areas, some setups must be connected to high (vertical) benchmarks with a leveling ruler and not a staff. In that case, three previous errors cannot be eliminated, but the staff must be calibrated to precisely determine the deviation of the baseplate from the nominal zero value. Thus, those errors can be minimized in post-processing by the application of zero error correction values. Error in the tension of invar tape, error of graduation lines, and error due to change in staff scale all have the same source: the invar tape. If the spring holding the tape changes its properties caused by temperature change or some mechanical influence, the tape will change the length, and thus the staff scale will change. If the graduation on the invar tape is incorrectly applied (the deviation of graduation lines is greater than the allowed 7 µm) the scale of the staff will be inconsistent or will differ from the nominal value of more than the allowed 20 ppm. Also, the staff scale can vary because of the change of length of the invar tape due to air temperature change or to daily exposure to mechanical forces. All these errors have a combined influence on the change of staff scale. For that reason, precise leveling staffs must be regularly calibrated to check for possible deviation of scale and, if there is any, to determine its exact value.

As said, to determine the leveling staff imperfections, such as zero error and change in scale, staffs must be calibrated. Calibration of a leveling staff or any analog linear measuring instrument (measuring tape or similar) is a procedure that has the goal to determine its true scale value and possible zero error. In that procedure, nominal values indicated on a scale and their measured values are compared. From that set of differences which are scattered discrete data points, it is needed to obtain a mathematical model. That model must be representative of the given data and simple and precise for later application in post-processing of the field leveling data. Praxis has shown that regression straight line is the best suited mathematical model for this purpose. In Figure 1, an example of calibration data for the right graduation of 3 m leveling staff is presented. Deviations of graduation lines are represented with orange points. The regression line and its equation in explicit form, *y* = *ax* + *b*, are shown. The slope coefficient *a* defines the direction of the regression line, which represents the *m*_0_—the coefficient of deviation between actual and nominal scale and is given in parts per million (ppm). Value *b* is *y* coordinate of the intersection of the regression line with the axis *Y* which represents the zero error of the leveling staff and is given in micrometers (10^−6^ m). In addition to these parameters, it is always necessary to have the mean air temperature during the calibration of the leveling staff, and later during field measurements air temperature paired with each staff reading. In that way, it is possible to compute the corrected reading value of the leveling staff (Equation (1)). The corrected value of leveling staff reading is computed according to the expression [20]: (1)$$L=L^{\prime }\{1+\left[m_{0}+\alpha_{t}\left(T-T_{0}\right)\right]\cdot 10^{-6}\},$$
where *L* is the corrected value of leveling staff reading (m), *L*′ is the leveling staff field reading, *m*_0_ is coefficient of deviation between actual and nominal scale (ppm), *α_t_* is the coefficient of thermal expansion of invar (µm/°C or ppm/°C), *T* is air temperature during field measurement (°C), and *T*_0_ is air temperature during calibration that should be 20 °C.

The theoretical basis for the automatic calibration of leveling staffs was established by Schlemmer [21]. In that thesis, a comparator that uses an electric motor guided trolley with the staff was elaborated. Graduations would be detected by a photoelectric microscope, while an optical interferometer would be used for measuring the staff displacement. Based on these instructions, three comparators were produced at three German universities: Physikalisch-Technische Bundesanstalt Braunschweig (PTB), Universität der Bundeswehr München (UniBwM), and Technische Universität München (TUM) [22]. With the development of calibration methods, new technologies were introduced, including cameras and algorithms for automatic edge detection. According to the publicly available literature, the first comparator modernized with a charge-coupled device (CCD) camera was at Eidgenössische Technische Hochschule (ETH) in Zurich [23]. It was constructed as an 8 m-long horizontal comparator with a guided trolley for the leveling staff; the displacement was measured using an interferometer HP 5519A; and the camera had a resolution of 8 μm. The achieved standard deviation from seven calibrations of one leveling rod was 5 μm. In his doctoral dissertation in 2000, Friede [24] described the theoretical basis for the development of a vertical and horizontal comparator with a CCD camera and interferometer with guided trolley for the leveling staff at the TUM. Due to issues with the functionality of the comparator, modifications were made to both the hardware and software components of the comparators in 2004. Using the vertical comparator, it is possible to determine the length of a leveling staff measuring 3 m with a measurement uncertainty of 1 ppm and to determine the zero error of the staff with an uncertainty of 6 μm. Meanwhile, the horizontal comparator, located in a temperature chamber, allows for the determination of the coefficient of thermal expansion of an invar tape with an uncertainty of 0.5 ppm [25]. Due to achieving the mentioned uncertainties, the Laboratory of the Chair of Geodesy at the TUM is considered the best laboratory in the field of research and development of leveling staff calibration systems and represents a reference point for all researchers in that field. For the calibration of precise leveling staffs at the Faculty of Civil Engineering and Geodesy (FGG) of the University of Ljubljana, the Zeiss MSGL001 comparator was used, one of only four such instruments in the world. It was an optical–mechanical instrument designed for objects up to three meters in length. In the year 2000, it was adapted for calibrating analog precise leveling staffs and measuring tapes. Measurements on such a device were time-consuming and demanding, and the appearance of coded precise leveling staffs was the final motivation for upgrading the comparator. The tracks of the existing comparator served as the basis for setting up the new positioning drive on which the optical system with a CCD camera is mounted. The positioning drive is connected to the Renishaw linear encoder reading head [26]. The uncertainty of determining the measurement of precise leveling staffs was achieved to be 1 ppm or even less [20]. At Stanford University, United States, in collaboration with TUG, a vertical comparator was built in 2003. For displacement measurement, it uses the Agilent (HP) 5517B laser interferometer with a resolution of 0.6 nm. The comparator is located in an old underground tunnel with concrete walls 1 m thick, which provides excellent stability of the air temperature, checked by six temperature sensors placed along the leveling staff’s path. Along with data from one measurement sensor for air humidity and one sensor for air pressure, the temperature data during calibration are used to compute the atmospheric correction of the displacement measured by the laser interferometer. As already said, the unique feature of this comparator is that it is designed to calibrate the leveling set or to calibrate only the leveling staff using the built-in CCD camera [5]. During the calibration of the leveling staff, images captured by the CCD camera are analyzed by software to detect the edges of the graduations. The position of the center of each graduation is determined with an uncertainty of 2.4 µm (k = 2). The scale of the precise leveling staff is determined with a measurement uncertainty of 1.2 ppm (k = 2) for a length of 3 m and 1.8 ppm for a length of 2 m [5,7]. In 2008, the existing vertical comparator at the Geodetic Metrology Laboratory of AGH University of Science and Technology in Krakow, Poland, was modified by replacing the Abbe microscope on a positioning drive with a CCD camera. The existing HP 5529A laser interferometer was retained as a length standard; the existing atmospheric sensors were also retained; and software support for managing and analyzing the calibration was developed. Through the calibration process using this comparator, a measurement uncertainty of 2 ppm was achieved in determining the scale of the leveling staff [27]. In addition to the vertical comparator, a horizontal comparator was also developed, which is located in a separate chamber where the air temperature is regulated from −10 °C to +50 °C, making it possible to determine the coefficient of linear thermal expansion of the invar tape [28]. In the literature, there can be found several more calibrators that use an electronic microscope or a CCD camera for capturing the images of the leveling staff: the comparator at the Department of Theoretical Geodesy of The Faculty of Civil Engineering of the Slovak University of Technology in Bratislava, Slovakia [29], the comparator at the Technical University in Ostrava, Czech Republic [29], the comparator at the Dresden Technical University in Germany [8], the comparator at the Geodetic Institute of University of Bonn (Uni Bonn) in Germany [30], and the comparator in the Photoelectric Measurement Technique Lab at the Xi’an University of Technology in China [31].

The development of the comparator for leveling staffs and measuring tapes at the Laboratory for Measurements and Measuring Technique (LMMT) of the Faculty of Geodesy at the University of Zagreb began in 1999 with the approval of the topic to Đuro Barković (then an assistant and currently a full professor) for his doctoral dissertation titled Automation of Comparation of Leveling Staves and Steel Tapes. As part of this work, a new calibration method was developed, and a completely new comparator was constructed, which became fully operational in 2002 when the doctoral dissertation was defended under a partially modified title Comparison of Leveling Staves by means of Sealed Linear Encoder [19,32]. Until 1999, the laboratory had a horizontal comparator mounted on the wall, which used an invar bar as the length standard. Due to observed shortcomings such as the limited number of reference points (only 11) over a length of 3 m, the high probability of parallax errors due to microscope sighting and reading, as well as the deformations and curvature observed on the comparator’s guide rails, it was decided to develop a completely new comparator based on their own design.

The comparator was assembled in several phases (Figure 2). In phase 1 three 4 m-long steel U-profiles were welded together to form the comparator body. In phase 2, on the left and the right upper body surfaces steel reinforcements and stainless-steel guide rails were mounted. For the calibration measurements to be reliable and accurate, the trolley with a microscope for observing the leveling staff graduations must move in a straight line without or with the least possible angular displacement around any axis (yaw, pitch, and roll). Therefore, the upper surfaces of the guide rails in the horizontal plane and the inner surfaces in the vertical plane were finely ground with an uncertainty of 0.01 mm/m, which was later confirmed by using geometric leveling. In phase 3, along the comparator bed at equal intervals seven clamps with clamping screws were mounted. Their purpose is to fixate the leveling staff in the middle of the comparator during the calibration. At the beginning of the bed, a half sphere was mounted on an aluminum stand. It imitates a spherical or a barrel shaped height benchmarks so that the staff baseplate always has the contact only at the same point. The comparator body was mounted on two steel frames. It was screwed on one frame, and it only rests on the second one to assure the dilatation of the material and the reduction of possible tension. After it had been determined that the comparator structure meets the planned requirements, a linear encoder Heidenhain LS 106C was mounted on the left inner side of the bed. Its expanded measurement uncertainty is *u*_95%_ = 0.2 µm + 0.3 ppm∙*L* [33]. A trolley was metal made construction with a mounted optical microscope, which rode on guide rails supported by eight ball bearings, of which four were used for moving horizontally on the guide rails, and the remaining four that spin around the vertical axis to push against the inside lateral sides of the guide rails. The trolley was connected to the encoder reading head. In that way when the trolley was moved backward or forward, the reading head would move with it, so it was possible to precisely determine the movement of the trolley.

After the comparator was assembled, software was developed. It consisted of two modules: measuring and computing. The calibration was conducted in a way that an operator using a microscope would aim at every graduation line edge three times. Nominal and measured values were recorded, and the difference was computed. From computed differences, a best fit model was computed, that is, a regression line. Then, a calibration report was produced in the form of a PDF document that contained the leveling staff data (product name, serial number, and type), comparation graph (discrete data and regression line), average air temperature, and calibration result: coefficient of deviation between the actual and nominal scale of the calibrated leveling staff. The achieved accuracy of determining the scale coefficient was 0.63 ppm [32]. Although accuracy was more than satisfying, the process of comparation measurement was time consuming and subjective to the operator’s influence. So, it was decided to develop a fully automated comparator by integrating a CCD camera into the existing comparator.

## 2. Construction of the New Comparator and Development of Calibration Method at the LMMT

For the construction and development of the new comparator, method, and corresponding software for the automatic calibration of geodetic linear scales, cooperation with the Chair of Machine Tools of the Faculty of Mechanical Engineering and Naval Architecture of the University of Zagreb (CMT-FMENA) was realized [34].

### 2.1. Hardware and Software Development

The first step in the design of the new comparator was to analyze the usability of the existing comparator basis and parts, and the conclusions were:The existing comparator is in a room with a stable temperature and equipped with a high-quality air conditioning system, ensuring a constant air temperature (deviation < 1 °C).The comparator is assembled and mounted twenty years ago, and all heavy mechanical parts are stable; therefore, the body of the comparator is extremely stable, and it is recommended to keep the entire body of the comparator.The guide rails for the guided trolley are well made, particularly concerning the high level of parallelism of the internal sides and coplanarity of the upper surfaces providing an excellent foundation for the upgrade.The existing linear encoder represents a stable standard of small measurement uncertainty, but to look in the future for an upgrade in a form of a laser interferometer.The existing guided trolley cannot support upgrades and needs to be replaced.The existing computer is outdated and did not meet the needs of the new method based on digital image processing.The existing software is not compatible with the new calibration method.

Based on the above, it was established that the construction of the existing comparator provides a stable foundation, and no extensive modifications concerning the comparator body are necessary. It can be the basis for the development of the new comparator components and the new calibration method. The first step towards development was a concept idea (Figure 3).

It was established that the new comparator should automatically acquire and analyze images and also determine the position of every graduation line edge. To accomplish that, at the CMT-FMENA, comparator parts were developed and constructed. Digital camera Imaging Source DMK 23U274 [35] with telecentric lens Sill Optics S5LPL2660/LED [36] was mounted on a servo-motorized positioning drive (Figure 4).

Also, a computer control system for control and management of positioning drive and camera, as well as for registration and processing of measurement data, was developed. In parallel with the development of the comparator and its control system, supporting software was also developed. Its functions are to manage the basic processes of the comparator’s control unit, as well as to control the positioning drive, capturing images, reading values from the linear encoder, running image processing and analysis algorithms, measuring data processing, and ultimately, generating calibration reports. The software was developed in Python and C/C++ programming languages, utilizing a range of libraries and support software, with the most important ones being OpenCV, LinuxCNC, GTK2, Numpy, Gstreamer, Matplotlib, LXML, PyGTK, and ReportLab. All the aforementioned libraries and tools fall under the domain of open source under the GPL (General Public License) license [37]. Most of the code can be easily altered or supplemented for testing with different settings or adding new linear scales that will be calibrated. Figure 5 presents a view of the comparator’s main graphical user interface.

After the initial test at the CMT-FMENA, all the mechanical components were integrated into a comparator (calibration) system at the LMMT (Figure 6). Comparator system software functionalities and detailed description of hardware components will be described in an appropriate article in the near future.

### 2.2. Process of Comparation and Mathematical-Statistics Background

During comparation measurements, software operates and monitors all the processes. Based on the input step, the positioning drive moves and stops while a photo of the visible part comparation item (scale) is taken. When the photo is taken, edge detection algorithms determine the position of the graduation line edge. In this software, edge detection is based on different algorithms such as the Gaussian filter, the Canny edge detector, the Sobel operator, morphological operations, and the Hough line transform, which are not the subject of this paper and will be also thoroughly described in an appropriate article. The basic data unit is a position of a graduation line edge $X_{0\left(i,n\right)}$ (in software data type *unmerged*) where *i* is the ordinal number of the graduation line edge, and *n* is the ordinal number of the same edge detection. Depending on the step size, every edge can be detected *N* times. For every detected edge from multiple measurements, an average value is computed (in software data type *merged* (index *m*)): (2)$$X_{m\left(i\right)}=\frac{1}{N}\sum_{n=1}^{N}X_{0\left(i,n\right).}$$

For every average position of the graduation line edge, the standard uncertainty of measurement is also determined: (3)$$s_{m\left(i\right)}=\sqrt{\frac{1}{N-1}{\sum_{n=1}^{N}\left(X_{0\left(i,n\right)}-X_{m\left(i\right)}\right)}^{2}}.$$

From the pair of one graduation line edges (lower (0) edge and (1) upper edge), the center of the graduation line is computed (in software data type *symmetry* (index *s*)): (4)$$X_{s\left(k\right)}=\frac{1}{2}(X_{m\left(i,0\right)}+X_{m\left(i,1\right)}).$$
where *k* is the ordinal number of the graduation line of the scale graduation that has the total amount of *K* graduation lines. Also, appropriate standard uncertainty is computed for each graduation line position: (5)$$s_{s\left(k\right)}=\sqrt{s_{m\left(i,0\right)}^{2}+s_{m\left(i,1\right)}^{2}}.$$

After each graduation (center) line position is computed, it is compared to its nominal position, which is a difference Δ*_k_* between the nominal value *X_Nom_* and the measured value of the graduation line: (6)$$\Delta_{k}=X_{Nom\left(k\right)}-X_{s\left(k\right)}.$$

Its standard uncertainty is equal to the standard uncertainty of the graduation line standard uncertainty *s_s_*_(*k*)_. As said in chapter 1 and can be seen from Figure 1, the comparation data graph is drawn by the application of nominal values on the X axis and corresponding Δ*_k_* on the Y axis. From that representation, approximate scale value can be estimated. To precisely determine zero error and scale deviation, the coefficient regression straight line must be modeled. In this case, regression straight line can be expressed in the following explicit form: (7)$$\Delta_{m}={\bar{m}}_{0}X_{s}+{\bar{X}}_{ZE}.$$
where Δ*_m_* are modeled values of differences between measured and nominal values of graduation lines, ${\bar{m}}_{0}$ is the adjusted value of the coefficient $m_{0}$, $X_{s}$ is the measured values of the graduation lines, and ${\bar{X}}_{ZE}$ is the adjusted value of the zero error. Regarding the fact that measurements are considered random variables with systematic influences and blunders eliminated and distributed by Gaussian distribution, adjustment is conducted by using the Gauss–Markov model with Least Squares Principle [38,39]. The estimated unknowns are computed by solving normal equations system ***Nx*** − ***n*** = 0, where: (8)$$N=A^{T}PA\;\mathrm{and}-n=-A^{T}Pl$$
in which ***N*** is the normal equations matrix, ***x*** is the deprived unknowns vector, ***A*** is the measurement matrix, ***P*** is the weight matrix, ***n*** is the absolute value of the normal equations vector, and ***l*** is the deprived measurements vector. The first step to solving it is to set the equations of corrections. In this case, where it is known that estimates are approximately zero, the expression is: (9)$$v_{k}=X_{s(k)}m_{0}+X_{ZE}-\Delta_{k}.$$
or in matrix form: (10)$$\overset{v}{\underset{K\times 1}{\left[\begin{array}{c} v_{1}\\ \vdots \\ v_{K} \end{array}\right]}}=\overset{A}{\underset{K\times 2}{\left[\begin{array}{cc} X_{s_{1}} & 1\\ \vdots & \vdots \\ X_{s_{K}} & 1 \end{array}\right]}}\overset{x}{\underset{2\times 1}{\left[\begin{array}{c} m_{0}\\ X_{NL} \end{array}\right]}}+\overset{-l}{\underset{K\times 1}{\left[\begin{array}{c} -\Delta_{1}\\ \vdots \\ -\Delta_{K} \end{array}\right]}}.$$

Since every graduation line position is determined with different accuracy, that is, has different measurement standard uncertainty, a stochastic component in the form of measurement weights is introduced into adjustment. The weight *p* of every single measurement of the *k*-th graduation line is computed by the following expression: (11)$$p_{k}=\frac{c}{s_{s(k)}^{2}}.$$
where *c* is the proportionality factor of constant value and $s_{s(k)}^{2}$ is the measurement estimate variance. Weight matrix ***P*** is then a diagonal matrix with *K* diagonal elements. Estimates of the unknowns are computed by the solution of the normal equations: (12)$$\underset{2\times 1}{x}={\underset{2\times 2}{N}}^{-1}\underset{2\times 1}{n}=\underset{2\times 2}{Q_{xx}}\underset{2\times 1}{n}.$$
where the covariance matrix of the estimated unknowns ***Q****_xx_* is the inverse value of the matrix ***N***. After adjustment, uncertainties of the estimated values ${\bar{m}}_{0}$ and ${\bar{X}}_{ZE}$ are given with standard uncertainties: (13)$$s_{\bar{m}\left(0\right)}=s_{0}\sqrt{q_{xx}}\;\mathrm{and}\;s_{ZE}=s_{0}\sqrt{q_{yy}},$$
where *q_xx_* and *q_yy_* are diagonal elements of the covariance matrix of the estimated unknowns, and the referent standard uncertainty *s*_0_ is given with the expression: (14)$$s_{0}=\sqrt{\frac{v^{t}Pv}{n_{f}}}$$
where *n_f_* is the number of supernumerary measurements and it is computed as *n_f_* = *n* − *u* where *n* is number of measurements and *u* is the number of the unknown values (in our case 2).

## 3. Testing the New Comparator at the LMMT

Once the comparator was fully assembled and mechanical and electronic tests were conducted, measurement tests were conducted [34].

### 3.1. Initial Testing of the Comparator

#### 3.1.1. Encoder Calibration

A calibration of the comparator, or more precisely, the comparator’s encoder, was conducted by the Precision Length Measurement Laboratory of the FMENA of the University of Zagreb, being also the National Laboratory for Length in the Republic of Croatia. The calibration measurement procedure was conducted by performing two repetitions of the positioning uncertainty measurement by using the Renishaw ML10 laser interferometer. The measurements were conducted in the positive direction of the comparator’s X-axis. Table 3 presents the calibration results, while Figure 7 shows a graph depicting the deviations of the comparator’s linear encoder compared to measurements obtained with a laser interferometer. The following findings were established:The modeled scale of the encoder was estimated to be 6.4 ppm based on weighted adjustment from two measurements.The zero error of the encoder was found to be 0.00 µm through repeated referencing of the positioning drive to the zero position, revealing differences of about 10 nanometers.The measurements exhibit a high level of repeatability, enabling accurate measurements with the application of calibration-based corrections.The uncertainty estimation of the comparator’s measurement deviations was determined from repeated measurements, yielding an average value of 0.81 µm, which will be considered in the computation of measurement uncertainty.A noticeable deviation of the measurements from the factory values was observed, indicating the recommendation for periodic annual or semi-annual calibration of the encoder to assess its stability or integrating a laser interferometer into the comparator.

#### 3.1.2. Transversal and Longitudinal Deviation of the Field of View of the Camera

After determining the correction value for longitudinal deviations due to imperfections in the comparator’s encoder through calibration, it was necessary to determine the transversal deviation of the camera’s field of view from the longitudinal axis of the comparator. It is caused by the lateral displacement and tilt of the camera due to imperfections of the guide rails. The transversal deviations were determined a priori and experimentally. Considering the guide rails were sanded with a precision of 0.01 mm/m and the working length of the comparator (3 m), a maximum height difference between the left and right guide in a transversal cross-section can be assumed to be 0.06 mm. Given the transversal spacing of the center of the bearings that vertically support the positioning drive on the guides (257 mm), there is a possible maximum transversal inclination of the positioning drive of 48.16 arc seconds. The distance between the camera and the object being calibrated is usually 363 mm, so considering the transversal inclination, the maximum transversal deviation of the camera’s field of view is 84.8 µm, which influences the longitudinal error of 0.004 µm, a negligible amount.

The transversal deviations were measured by stretching a thin dark thread with a thickness of 0.16 mm across the entire calibration range of the comparator. A clean white tape was placed below the thread to enhance contrast. The start and end points of the thread were positioned in the center of the camera’s field of view, and images were captured with a positioning drive step of 1 mm from the beginning to the end (0–2999 mm) of the comparator. The analysis of the measurement results was performed using a program developed in the Matlab software package. Deviations of the center of the camera’s field of view relative to the thread passing through the center of the field of view at the beginning and end of the comparator system were determined. The obtained data were then transformed into a coordinate system in which the abscissa passes through the center of the field of view at the beginning (0 mm) and end (2999 mm) of the comparator system, representing the longitudinal axis of the comparator (Figure 8). Numerical analysis revealed a maximum transversal deviation of 72 µm in absolute value, which is consistent with the a priori estimation. By applying the Pythagorean theorem, a maximum longitudinal deviation due to transversal deviation was determined to be 0.004 µm in absolute value, which represents a negligible influence on the final measurement uncertainty and does not need to be considered in the computation of the measurement uncertainty budget.

Since it was not possible to experimentally determine the longitudinal deviation of the camera’s field of view, an a priori estimation was made. The distance between the front and rear bearings on which the drive vertically rests on the guides is 370 mm, which, considering the uncertainty in the grinding of the rail guides (0.01 mm/m), results in a possible maximum longitudinal inclination of the positioning drive of 2.06″. Analogous to the computation of transversal deviation, the estimated value of the longitudinal deviation of the camera’s field of view is 3.6 µm. This value needs to be considered in the computation of measurement uncertainty budget.

#### 3.1.3. Determination of the Comparator Zero

When the precise leveling staff is mounted for calibration in the comparator, the base plate is pressed against a half sphere. The bottom edge of the baseplate cannot be captured by the camera, meaning the software cannot detect it because it is rounded and out of focus. To reliably determine the zero error of the leveling staff, a special method for determining the zero position of the comparator had to be developed. The zero position of the comparator is the position when the encoder is homed and the origin coincides with the machine zero, while the center of the camera’s field of view is at the 0.000 mm position. At that point, the central transversal index (equivalent to the horizontal thread of the level’s crosshair) must coincide with the bottom of the reference surface, i.e., the lower edge of the baseplate. As a result, a special steel standard (Figure 9) was created, featuring finely and sharply sanded transversal and longitudinal square ridges. The upper surface of the ridges is at the same height as the invar tape during calibration to avoid camera elevation or lowering due to focusing. The lower surface of the standard is painted in a black matte color to achieve high contrast and enable the detection of the edge of the ridge as the graduation line edge.

The distance between the base of the standard and the transversal ridge was determined using a three-coordinate (CMM) measuring device at the aforementioned Laboratory for Precision Length Measurements. That distance is taken as a quasi-true value. In the next step, the edge of the transversal ridge was determined using the comparator from three measurements. The obtained value was compared to the quasi-true value, and the difference between these two values was computed with an uncertainty of 12 µm. Within the comparator software, the zero position was adjusted based on the computed difference. Such calibration of the zero position of the comparator needs to be performed periodically due to mechanical wear of the comparator’s reference, which results in the displacement of the zero position.

### 3.2. Testing the Calibration Data and Results

#### 3.2.1. Normality and Homogeneity Tests

As said in Section 2.2, data processing is conducted, and the results are given regarding the fact that measurements are considered random variables with systematic influences and blunders eliminated and distributed by Gaussian distribution. To verify that, a series of statistical tests were conducted. For that, two pairs of 3 m leveling staffs were used: one pair of precise leveling staffs with a centimeter graduation and one pair of precise leveling staffs with a coded graduation. To test normal distribution, measuring samples were taken. At graduation lines 40 mm, 1510 mm, and 2960 mm, both edges were measured with a sample size of 146 measurements, that is, with a positioning drive step of 0.1 mm. Then, two tests were applied: Pearson’s chi-squared [38] and D’Agostino–Pearson [40,41] normality tests. Both test results were positive, and it was concluded that the data are distributed normally.

After confirming that the measurements taken on the comparator are normally distributed, homogeneity tests were performed to determine if all measurements have equal standard measurement uncertainty, expressed as the standard uncertainty of the measurements. Bartlett’s [42] and Cochran’s [43] tests were conducted on an identical set of six measurements as normality tests. Both tests confirmed the homogeneity of the measurements from which the samples were taken. Therefore, it is assumed that throughout the entire working range of the comparator, all measurements are of equal accuracy.

#### 3.2.2. Determination of the Positioning Drive Optimal Step

After establishing that the measurements are normally distributed and homogeneous, the next step was to determine if the measurement results could be achieved with fewer redundant measurements without increasing the measurement uncertainty. Calibrations were performed with different positioning drive steps, and for each calibration, the comparation time, the number of measurements of the graduation line edge, and the average standard uncertainty of graduation line edge measurement of the right and left graduation line edges were determined (Table 4). Initially, it was concluded that for all scanning steps, the uncertainties were approximately equal, except for a slight increase observed for a step of 5.0 mm. Therefore, the possibility of a 5.0 mm step was eliminated. The measurement values obtained with steps of 1.0 mm and 2.0 mm were compared next. Both values resulted in approximately equal standard measurement uncertainties for determining the division line edge. However, when using a step of 2.0 mm, a constant difference was noticed in the number of measurements between the left and right graduation. To avoid this, it was decided to perform all calibrations with a step of 1.0 mm.

All subsequent measurements were taken at the same positions with a scanning step of 1.0 mm, meaning each graduation line edge was measured 14 times. Average edge position values with corresponding standard uncertainties were determined, and the homogeneity of measurements was tested, with positive results. These average values were then compared to the average values obtained from 146 repeated measurements. For this purpose, tests comparing the average values were conducted to determine if the average values obtained from a smaller number (*n* = 14) of measurements had the same value as those obtained from a larger (*n* = 146) number of measurements. T-tests (Student’s *t*-tests) [38] have shown that the results obtained with only 14 measurements can be considered identical to the results obtained with 146 measurements.

#### 3.2.3. Testing the Repeatability

After proving that the same results can be achieved with a reduced number of measurements, it was needed to test the repeatability of the obtained measurements. To test the repeatability of calibration, specifically the repeatability of determining the geodetic linear scale value, two pairs of precise leveling staffs that were used for all previous measurements were calibrated in two repetitions on two different days with repositioning in the working position in the comparator. All measurements were corrected for the encoder scale deviation, and adjustment was performed. Table 5 presents the results of the two independent calibrations for the two pairs of leveling staffs. The adjusted values of the coefficient of scale and the zero error from the two calibrations were compared using a T-test for two independent adjusted values of the unknowns. Tests have shown that there is no significant difference in results between repeated calibrations.

## 4. Uncertainty Assessment of the New Comparator at the LMMT

### 4.1. A Priori Inner Uncertainty Assessment

An a priori estimation of measurement uncertainty includes all possible influences when measuring the position of a single graduation line edge, which have been modeled or empirically defined earlier in this article:Measurement uncertainty of the comparator encoder:
$$u_{encoder}=0.1\;µm+0.15\;\mathrm{ppm}$$

Measurement uncertainty in estimating the scale error of the comparator encoder:


$$u_{encoder\;scale}=0.8\;µm$$


Measurement uncertainty in detecting the graduation line edge:

Numerous studies have addressed the estimation of measurement uncertainty in edge detection on images. However, most of them focus on visual analysis, comparing the original image with the obtained results and focusing on the quantity of correctly detected lines. In certain studies [44,45,46,47], the estimated uncertainty in using the Hough transform, which is used for the final detection of graduation line edges in the developed comparator software, ranges from 0.1 to 0.5 pixels. Since most of these estimates are based on comparisons with artificially created images with added noise, the upper value of 0.5 pixels will be used for estimating the measurement uncertainty of the edge detection algorithm. The resolution of the image acquisition system is 13.03 µm, so the measurement uncertainty of the edge detection algorithm is: $$u_{algorithm}=6.5\;µm$$

Transversal deviation of the camera’s field of view center from the comparator longitudinal axis:


$$u_{transverse\;fov}=3.6\;µm$$


The overall a priori estimation of measurement uncertainty for a single measurement of an individual graduation line edge with a probability of 68.3% (k = 1) is: $$u_{edge}=\sqrt{u_{encoder}^{2}+u_{encoder\;scale}^{2}+u_{algorithm}^{2}+u_{transverse\;fov}^{2}}=7.5\;\mu m\;+\;0.15\;\mathrm{ppm},$$
and expanded uncertainty with a probability of 95% (k = 2) is:$$U_{edge}=14.7\;µm+0.30\;\mathrm{ppm},$$

The a priori estimation of measurement uncertainty for the center of the graduation line based on individual measurements of the upper and lower edges is:

$u_{line}$ = 10.6 µm + 0.2 ppm with a probability of 68.3%, and the expanded measurement uncertainty is:

$U_{line}$ = 20.8 µm + 0.4 ppm µm with a probability of 95%.

### 4.2. Experimental Inner Uncertainty Assessment

Each graduation line edge was measured 14 times during the calibration. From measured edges, a graduation line center was averaged, and measurement uncertainty $u_{m\left(k\right)}$ was computed. The computed measurement uncertainty was summed with the uncertainty of the comparator encoder ($u_{encoder\;scale}$ = 0.8 µm). The measurement uncertainty for each graduation line was computed according to the equation:(15)$$u_{gl\left(k\right)}=\sqrt{u_{s\left(k\right)}^{2}+u_{encoder\;scale}^{2}}$$

The mean measurement uncertainty of the graduation line of one graduation of the geodetic linear scale was computed as the root mean square all of the graduation lines uncertainties in one graduation:(16)$$u_{graduation}^{2}=\frac{1}{K}\sum_{k=1}^{K}u_{gl\left(k\right)}^{2}$$

The average standard measurement uncertainty of the graduation lines for all measurements on the comparator, obtained using the new method, was computed from all performed measurements analogously to Equation (16). Table 6 provides the standard measurement uncertainties for the calibration of the leveling staffs in two independent repetitions, as well as the final standard measurement uncertainty for the measurements conducted using the new method on the comparator. From the table, it can be observed that the standard measurement uncertainties of the graduation lines are lower than the estimated measurement uncertainty, which was achieved through redundant measurements of each graduation line edge.

### 4.3. External Comparison

For assessment of calibration results, an interlaboratory comparison was organized in cooperation with the Laboratory of the Chair of Geodesy at the TUM. A 3 m coded staff that only serves as a testing artifact was calibrated on the new comparator at the LMMT. Then, it was sent to Germany to be calibrated. After the return, it was once again calibrated at the LMMT. The calibration results from calibration certificates are given in Table 7. It can be concluded that the new comparator with the new method gives approximately the same results as the TUM comparator that is considered the best in the calibration of leveling staffs.

## 5. Discussion

Metrology laboratories that hold national standards for measurements are the foundation of stability and reliability in measurement systems. Specifically, metrology laboratories with an established quality system for measurement traceability ensure the homogeneity of measurements at the national level. For precise geodetic measurements, standards for angles and particularly for length are of vital importance to ensure the homogeneity of national coordinate systems. To achieve this, it is necessary to carry out periodic calibrations of geodetic instruments and equipment, including all types of geodetic measuring devices. One of the steps is the development of new calibration methods and calibration systems. At the LMMT, a new geodetic linear scale calibration system, with a special focus on precise leveling staffs, has been developed. In the beginning, the main goal was to develop a system that would:Provide accurate and precise results comparable with other worldwide calibration systems.Be easy to upgrade, hardware- and software-wise.Be budget-friendly.

After the hardware assembly and software development in academic cooperation with the Chair of Machine Tools of the FMENA, it was concluded that the calibration system is an open system that can be easily upgraded. Also, when compared to some other known budgets in the development of such systems, the costs were minimal and acceptable for the existing budget. In the next steps, extensive research was done with the following results:The a priori inner uncertainty for a single measurement of an individual graduation line edge is:
$$U_{edge}=14.7\;µm+0.30\;\mathrm{ppm},$$

The a priori inner uncertainty for a single measurement of an individual graduation line center is:


$$U_{line}=20.8\;µm+0.4\;\mathrm{ppm},$$


Mean experimental inner uncertainty of an individual graduation line center for the whole graduation is:


$$U_{comparator}=14.13\;µm,$$


External comparison has shown that calibration results are not significantly different when compared to calibration results of a reliable calibration system.

Numerical indicators show that the new calibration system has a slightly bigger measurement uncertainty for the single measurements which is then compensated by supernumerary measurements. Calibration results are within the uncertainty of other confirmed calibration results. The conclusion is that the new calibration system passes all the tests and can be freely used.

Regarding the observed deviations in the encoder data, to maintain or even reduce the measurement uncertainty of the calibration system and increase measurement reliability, a possible future upgrade could involve replacing the existing linear encoder. One solution that presents itself is the use of a laser interferometer, which has a measurement uncertainty of less than 0.5 ppm. Additionally, the laser interferometer could be applied in the development of other calibration methods and devices in the laboratory.

## Figures and Tables

**Figure 1 sensors-23-06358-f001:**
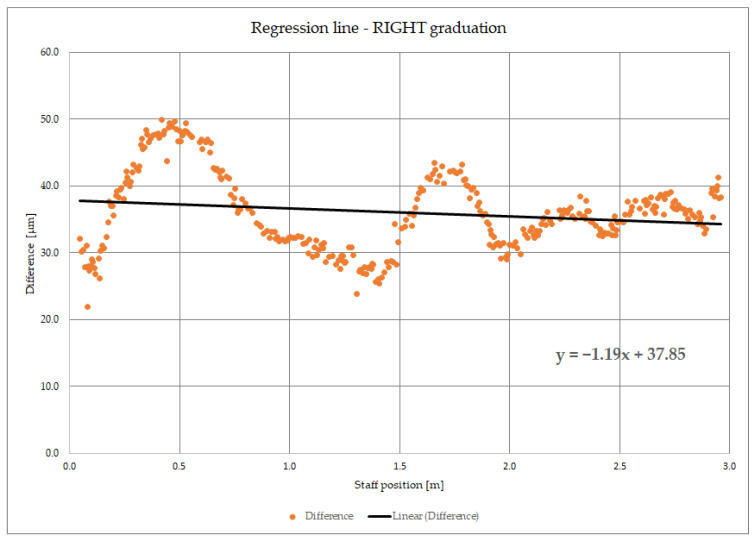
Example of calibration data graph for the right graduation of leveling staff Wild GPLE 3N (3 m analog). Orange points—differences of nominal and measured position of graduation lines, black line—regression line of differences.

**Figure 2 sensors-23-06358-f002:**
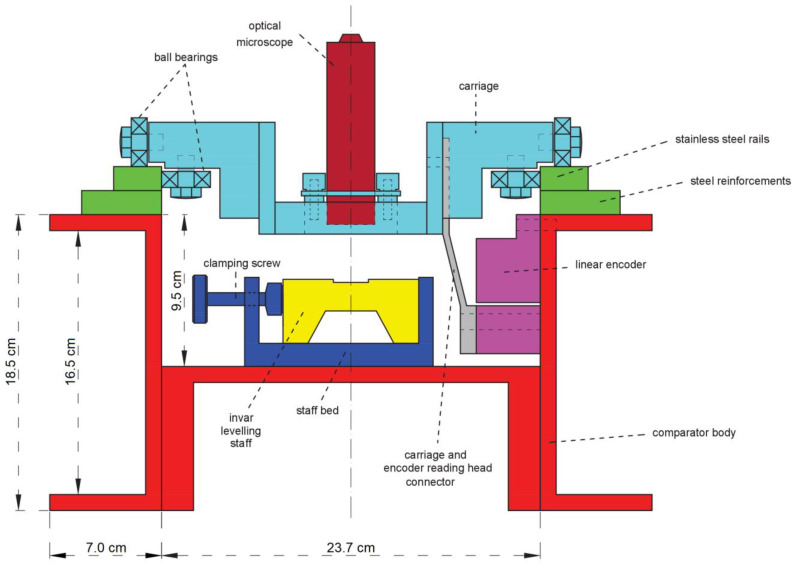
Cross section of a previous LMMT comparator. Colors represent parts that were assembled in several phases: 1. steel comparator body (red), 2. steel reinforcements and stainless-steel guide rails (green), 3. staff clamp with clamping screw (blue), 4. linear encoder (magenta), 5. trolley (cyan) connected to encoder and equipped with microscope (dark red).

**Figure 3 sensors-23-06358-f003:**
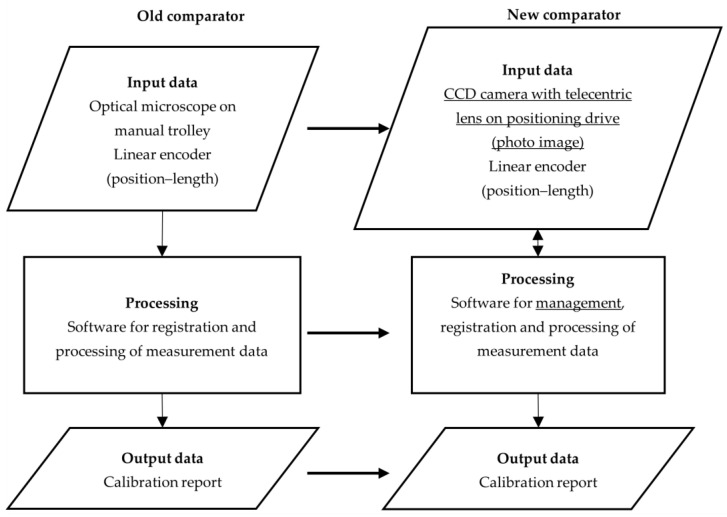
Conceptual representation of the old and the new comparators with planned hardware upgrades.

**Figure 4 sensors-23-06358-f004:**
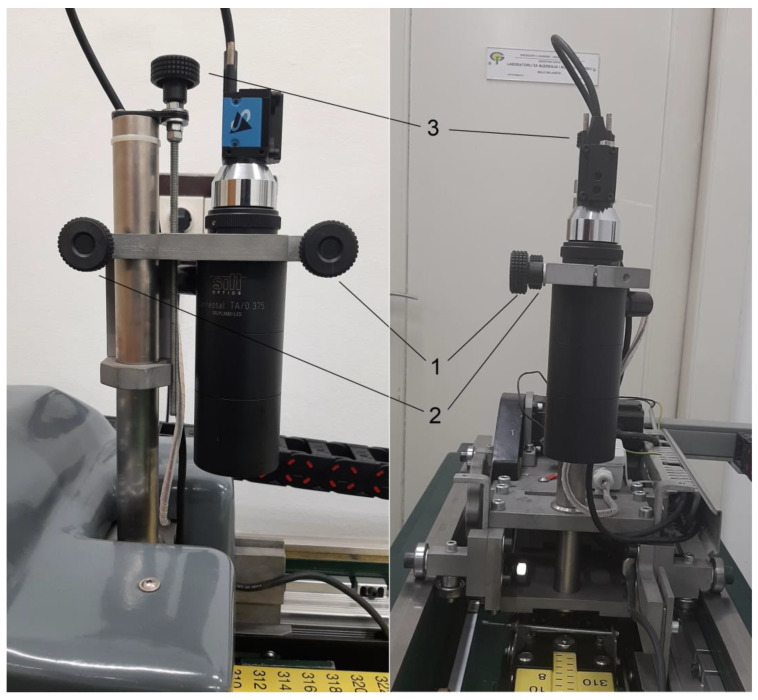
Digital camera with telecentric lens mounted on a positioning drive: lens and camera carrier screw (1), positioning screw for lens and camera carrier (2), and the screw for fine movement of the lens and camera carrier in the vertical direction (3).

**Figure 5 sensors-23-06358-f005:**
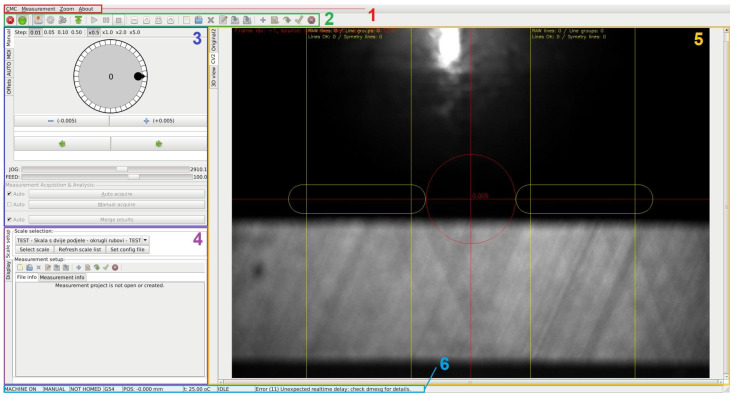
View of the software main graphical user interface that consists of: (1) Menu bar, (2) Toolbar, (3) Positioning drive management window, (4) Calibration measurement and graphics management window, (5) Graphics and data window, and (6) Status bar.

**Figure 6 sensors-23-06358-f006:**
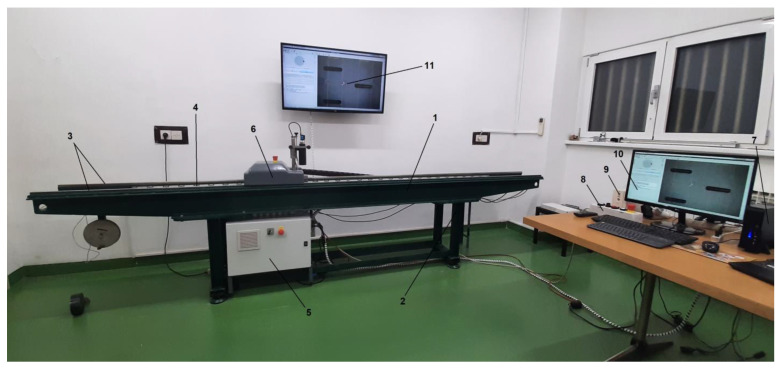
Fully assembled new comparator at the LMMT: (1) body and bed, (2) stand, (3) guide rails, (4) linear encoder, (5) computer control system, (6) positioning drive with lens and camera, (7) data storage computer, (8) remote controller, (9) Uninterruptible Power Supply, and (10 and 11) monitoring screens.

**Figure 7 sensors-23-06358-f007:**
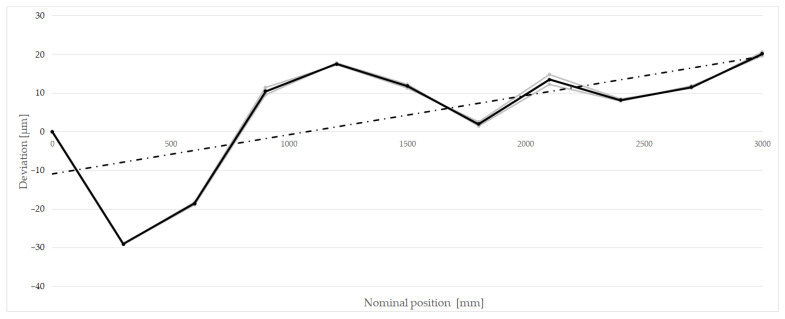
Calibration graph of the comparator’s encoder (black points and lines—average deviations, dash-dotted line—regression line).

**Figure 8 sensors-23-06358-f008:**
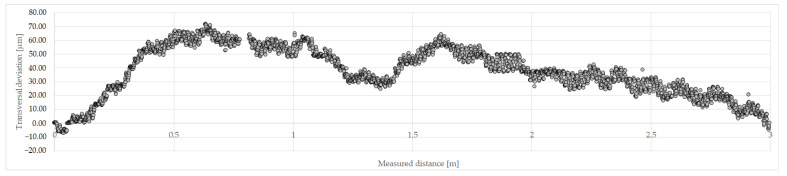
Transversal deviations graph of camera’s field of view center from the comparator longitudinal axis.

**Figure 9 sensors-23-06358-f009:**
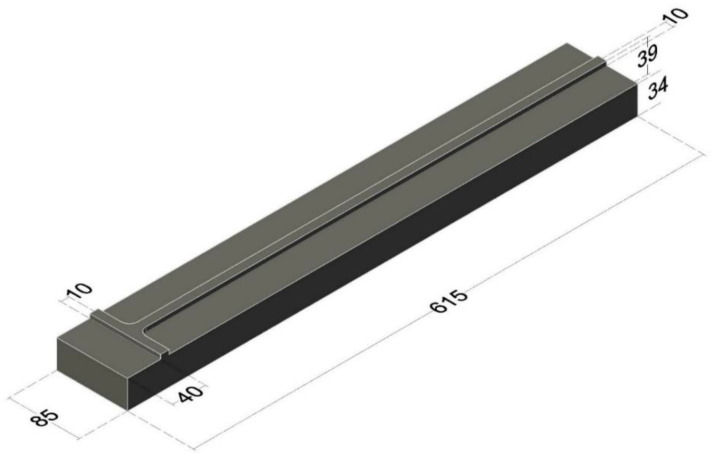
Standard for determining the difference in the zero position of the comparator (approximate dimensions given in mm).

**Table 1 sensors-23-06358-t001:** Comparator types by construction characteristics [4].

Construction Characteristic	Type	Advantages	Disadvantages
Object of comparation	Leveling staff	✓ Smaller uncertainty of comparation	✗ No systematic influence of level on comparation results
	Leveling set	✓ Systematic influence of level on comparation results	✗ Bigger uncertainty of comparation
Leveling staff position	Horizontal	✓ Simpler comparator construction	✗ Staff is not calibrated in a field working position
	Vertical	✓ Staff is calibrated in a field working position	✗ More complex and expensive construction
Leveling staff reading	Analog (optical microscope or optical level)	✓ More accurate detection of damaged graduation line edges	✗ Susceptible to operator’s subjectivity ✗ Longevity of comparator measurement
	Digital (camera or digital level)	✓ Operator’s subjectivity eliminated ✓ Speed	✗ Possible data redundancy
Distance (movement) measurement device	Linear encoder	✓ Simplicity ✓ Relatively low price	✗ Bigger uncertainty
	Laser interferometer	✓ Small uncertainty	✗ Working complexity ✗ Subjected to atmospheric conditions
Moving part	Leveling staff	✓ Fixed reading device (optical axis always with fixed orientation)	✗ Comparator must be at least twice as long as the leveling staff
	Reading device	✓ Smaller construction of the comparator	✗ Possible movements of reading device axis

**Table 2 sensors-23-06358-t002:** Overview of systematic errors of precision leveling staffs [4,18,19].

Systematic Error	Cause	Elimination/Minimization
Error due bending and torsion of staff	Deformation of staff body depending on air temperature and humidity	Eliminated by production technology (never use bent staff)
Error due non-verticality of staff	Maladjustment of spot bubble (circular bubble level)	Elimination by testing and adjustment
Error due non-flatness of baseplate	Baseplate is uneven (bumpy) or is not perpendicular with respect to staff (vertical) axis	Minimization by calibration or elimination by measurement method (use of baseplate ring)
Zero error of one staff	Bottom plane of baseplate does not coincide with zero graduation line	Minimization by calibration or elimination by measurement method
Zero error for a pair of staffs	Baseplates do not refer to the same nominal zero graduation line	Minimization by calibration or elimination by measurement method
Error in tension of invar tape	Change in tension force of invar tape due to mechanical influences	Minimization by calibration or adjustment
Error of graduation lines	Faulty application of graduation lines on invar tape	Minimization by production technology and calibration
Error due change in staff scale	Variation of invar tape length due to temperature change and day-to-day variation at 20 °C	Minimization by calibration and application of corrections

**Table 3 sensors-23-06358-t003:** The calibration results of positioning in the positive direction of the comparator X axis.

Nominal Position	Deviation			
	1st Measurement	2nd Measurement	Average	Standard Uncertainty
[mm]	[µm]	[µm]	[µm]	[µm]
0	0.00	0.00	0.00	0.00
300	–29.00	–29.10	–29.05	0.07
600	–18.90	–18.20	–18.55	0.50
900	9.50	11.40	10.45	1.34
1200	17.80	17.30	17.55	0.35
1500	12.30	11.30	11.80	0.71
1800	1.50	2.50	2.00	0.71
2100	12.20	14.80	13.50	1.84
2400	7.90	8.50	8.20	0.42
2700	11.80	11.30	11.55	0.35
2999	19.70	20.60	20.15	0.64
			u_calibration_=	0.81

**Table 4 sensors-23-06358-t004:** Comparison of comparation results with different positioning drive (camera) step.

Step [mm]	0.1	0.2	0.5	1.0	2.0	5.0
Comparation time (3 m)	≈24 h	≈12 h	≈4.5 h	≈2 h	≈1 h	≈0.5 h
Graduation line edge detections (measurements)	146	73	29/30	14	7/8	3
Average standard uncertainty of graduation line edge [µm]	6.0/5.0	6.1/5.0	6.0/5.0	5.9/5.1	5.8/5.2	6.4/5.3

**Table 5 sensors-23-06358-t005:** Results of calibration of two pairs of precise leveling staffs with corresponding uncertainties.

Leveling Staff (Type)	Serial Number	Calibration Order	Right Graduation Coded Graduation				Left Graduation			
			$m_{0}$	$s_{\bar{m}\left(0\right)}$	${\bar{X}}_{ZE}$	$s_{ZE}$	$m_{0}$	$s_{\bar{m}\left(0\right)}$	${\bar{X}}_{ZE}$	$s_{ZE}$
			[ppm]	[ppm]	[µm]	[µm]	[ppm]	[ppm]	[µm]	[µm]
Wild GPLE 3N (double graduation, 1 cm)	20,840	1st	–2.28	0.44	–6.6	0.7	–0.89	0.48	–1.3	0.8
		2nd	–2.62	0.43	–5.6	0.7	–1.31	0.47	–0.4	0.8
	20,842	1st	3.14	0.40	2.4	0.7	2.85	0.43	10.9	0.7
		2nd	3.10	0.42	2.7	0.7	2.75	0.47	10.7	0.8
Wild GPCL3 (coded graduation)	27,646	1st	1.92	0.40	30.7	0.7				
		2nd	2.65	0.39	30.3	0.6				
	27,648	1st	0.75	0.35	37.9	0.6				
		2nd	0.58	0.37	38.1	0.6				

**Table 6 sensors-23-06358-t006:** Standard measurement uncertainties of graduation lines centers.

Leveling Staff (Type)	Serial Number	Calibration Order	Graduation	$\mathrm{Standard}\;\mathrm{Uncertainty}\;u_{graduation}$
				[µm]
Wild GPLE 3N (double graduation, 1 cm)	20840	1st	Right	7.37
			Left	6.52
		2nd	Right	7.30
			Left	6.45
	20842	1st	Right	7.85
			Left	6.31
		2nd	Right	7.84
			Left	6.46
Wild GPCL3 (coded graduation)	27646	1st	Coded	7.69
		2nd	Coded	7.48
	27648	1st	Coded	7.55
		2nd	Coded	7.43
			$u_{comparator}$=	7.21
			$U_{comparator}$=	14.13

**Table 7 sensors-23-06358-t007:** Calibration results from the interlaboratory comparison.

Leveling Staff (Type)	Serial Number	Laboratory	$m_{0}$	$s_{m\left(0\right)}$	${\bar{X}}_{ZE}$	$s_{ZE}$
			[ppm]	[ppm]	[µm]	[µm]
Wild GPCL3 (coded graduation)	25557	LMMT	−4.65	0.26	−99	1
		TUM	−4.78	1.15	−96	2
		LMMT	−4.57	0.29	−98	1

## Data Availability

The data presented in this study are available on request from the corresponding authors.

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
