# Peer review of "Measuring Uncertainty Analysis of the New Leveling Staff Calibration System"

_sensors, 2023, doi:10.3390/s23146358_

Round 1

Reviewer 1 Report

This paper developed a set of hardware and software for assessment of the measurement uncertainty of determining the position of the edges of the graduation lines and determining the scale of precise centimeter and coded levelling staffs. The experiment results indicate the effectiveness of the equipment. However, there are some suggestions for the authors.

(1) The matrix in the formula should be italicized, such as formula (8), (12) and (14).

(2) Environmental parameters such as experimental temperature need to be explained in detail.

(3) Software functional modules and computer requirements need to be explained in detail.

Reviewer 2 Report

The article is interesting and timely. It deals with improving the comparator of leveling rods and increasing its accuracy, including uncertainty analysis. Theoretical assumptions they are verified practically based on the statistics of own repeated measurements and by means of interlaboratory comparison.

The article is written legibly, concisely and without unnecessary sentences.

Chapter 1 is a very useful introduction to the issue of leveling rods and their calibration.

However, I am missing some in the list of laboratories that perform calibration of leveling rods, which are also state metrology laboratories for length.

Minor typos and corrections:

In the formulas (2) - (4), the indices of the lower indexes change, which is not described and makes the matter incomprehensible. It would be good to improve it.

Formula (8): use "T" instead of "t" for the transposed matrix

Formula (14): it would be appropriate to state how n_f is calculated

I consider English technical terms to be a problem - e.g. staff/staves instead of rod and lines 288 and 652: Department of Machine ... instead of Chair of Machine...

line 104: redundant bracket - bubble level). In...

Table 2: 2 column, 3 row: Misadjustment instead of Maladjustment

line 152: wrong unit - should be ...ppm/°C), ...

Figure 7: in the description: there are more black lines in the picture, the regression line is a striped black line

I identified 4 selfcitations out of a total of 47 citations. Their introduction is justified and is not an obstacle.
